# Effect of Dry Aging of Pork on Microbiological Quality and Instrumental Characteristics

**DOI:** 10.3390/foods13193037

**Published:** 2024-09-25

**Authors:** Helena Veselá, Josef Kameník, Marta Dušková, František Ježek, Hana Svobodová

**Affiliations:** Department of Animal Origin Food & Gastronomic Sciences, Faculty of Veterinary Hygiene and Ecology, University of Veterinary Sciences Brno, 612 42 Brno-Královo Pole, Czech Republic; veselah@vfu.cz (H.V.); kamenikj@vfu.cz (J.K.); fjezek@vfu.cz (F.J.); h24012@vfu.cz (H.S.)

**Keywords:** total viable psychrotrophic count, *Enterobacteriaceae*, psychrotrophic lactic acid bacteria, *Pseudomonas* spp., color, hardness

## Abstract

Meat aging is an important process that affects the quality of meat and is traditionally used mainly for beef. However, in recent years, there has been an increasing demand for pork products subjected to dry aging. The aim of this study was to compare selected parameters (microbiological quality, instrumental analyses of texture and color of meat, *weight loss*) of pork neck and loin with bone and skin together subjected to dry aging for 14 days. The microbiological profile (total viable psychrotrophic count, *Enterobacteriaceae*, psychrotrophic lactic acid bacteria, *Pseudomonas* spp.) on the surface of the meat with the skin and the lateral cutting surfaces without skin was compared on the first day after slaughter and after 14 days of dry aging. The results of this study demonstrated that dry aging did not significantly deteriorate the microbiological profile. Statistically significant *weight loss*es were observed after 14 days of aging. The dry aging of pork had no significant effect on lightness (L*), redness (a*), and shear force. Significant differences were observed for yellowness (b*) and meat hardness (*p* < 0.05).

## 1. Introduction

The quality of pork is generally influenced by a variety of factors, such as the sex of the animal (which may result in the presence of boar taint), the fatty acid composition, the proportion and solubility of intramuscular connective tissue, the myofibrillar structure, the proportion of intramuscular fat, and the rate and extent of the decline in pH values after slaughter [1]. Consumers assess the sensory characteristics of the meat, which are referred to as “eating quality” [2]. In the initial stages of the assessment, consumers primarily consider the aroma, taste, juiciness, and tenderness of the meat. Tenderness is often considered the most important sensory attribute influencing customer satisfaction, as evidenced by several studies [3,4,5].

Meat aging is an additional factor and important influence on the quality that is part of the production process, which takes place in the processing stage at slaughterhouses or cutting plants [6]. Meat aging is a complex process to which groups of various endogenic proteases contribute, and this process begins immediately after the slaughter of the animal [7]. The structural integrity of myofibrils changes as a consequence of the degradation of muscle proteins such as titin, nebulin, and desmin [8]. In general, there are two primary methods of meat aging—wet aging and dry aging—and the former is used predominantly for beef. The two methods differ in the conditions and resulting quality. Wet aging is a process that was first introduced in the 1970s. It involves vacuum packing to protect meat from spoilage and drying out when refrigerated for 3 to 83 days [9]. The combination of dry and wet aging has also been tested [6]. But Vilella et al. [10] stated that combined dry and wet aging of beef had no significant differences in tenderness compared with wet and dry aging alone. However, dry-aged and combined-aged samples showed a greater total *weight loss* than wet-aged meat. Another combination of dry and wet aging is dry aging in aging bags that are highly permeable to moisture [11].

As in the case of beef, pork aging affects meat tenderness [12]. In the literature, a positive effect has been noted on pork aged for 4, 7, or 10–12 days compared to 1–2 days. Nevertheless, no positive effect has been demonstrated for post-mortem meat aging lasting 2 or 7 days, as 7 days may be insufficient [13]. Aaslyng et al. allowed pork loins to age at 4 °C for 2, 5, 7, or 10 days. The tenderness of the meat increased significantly with an aging period of up to 10 days (*p* = 0.001). There was no statistically significant difference between the sexes [14].

In the European meat processing industry, the typical duration reported for wet aging of pork is between two and six days [15]. Longer wet aging times are only possible at very low temperatures of around 0 °C and under strict hygienic conditions. However, according to some meat processors, only dry aging can be used for pork to improve its sensory characteristics (personal communication). Dry aging can be employed for up to 28 days in the case of pork, provided that the necessary conditions are met [16].

The aim of this study was to assess the effect of 14-day dry aging on pork meat quality by monitoring *weight loss*, microbiological quality, and instrumental parameters of meat tenderness and meat color.

## 2. Materials and Methods

### 2.1. Samples

Per agreement with a meat processor with its own commercial slaughterhouse, 15 pieces of pork neck and loin with bone and skin together were prepared from pigs slaughtered on site. The meat came from pigs (approximately 50% gilts and 50% barrows; large white × landrace) slaughtered at a single slaughterhouse within 50 km of the University of Veterinary Sciences Brno. An average slaughter weight of live pigs of 113.4 kg and an average weight of carcasses of 88.2 kg were recorded. The meat—specifically the pork neck and loin from the first cervical vertebra to the penultimate lumbar vertebra—was cut out with the skin intact. The samples of meat were prepared from the left halves of carcasses 24 h after slaughter from animals with average weight. The animals were initially stunned by applying an electric current (1.6 A), followed by bleeding in the recumbent position and scalding. Each piece of pork neck and loin with bone and skin together represented a single animal. The meat samples were then subjected to dry aging in an aging chamber for two weeks at 0–1 °C with an airflow of 0.5–1 m/s and relative air humidity of 80–82%. The sampling was carried out during April and May 2023.

### 2.2. Microbiological Parameters

#### 2.2.1. Sampling for Microbiological Examination

Samples were taken by swabbing the meat surface with an EZ Reach™ abrasive sponge with a handle (Bioing, s.r.o., Ivančice, Czech Republic) on the first day after slaughter (A, day 0) and following 14 days of dry aging (B, day 14). Two skin swabs were taken from the neck region and the loin region of the meat sample, each from an area of 100 cm^2^. Two swabs were also taken from the cut areas, one from the neck at the level of the first cervical vertebra and one from the loin at the level of the penultimate lumbar vertebra, each from an area of 25 cm^2^ (Figure 1). Once the samples for day 0 were thus taken, the meat was placed on racks and subjected to dry aging. All samples were transported to the laboratory under refrigeration conditions (4 °C). Analyses were conducted within three hours of sample collection. After 14 days of aging, samples were collected in the same manner but from different locations than those sampled on day 0. A total of 120 samples were collected for microbiological examination.

To ascertain the microbial quality inside the meat, four pork necks and loins with bone and skin together (prepared from the right halves of the slaughtered carcasses, the left halves of which were subjected to the aging procedure) were transported on day 0 to the laboratory under refrigeration conditions (4 °C). Following the application of a scorching gun to the surface of the samples, 10 g of meat from inside the neck and 10 g of meat from inside the loin were removed using sterile instruments for microbiological analysis (n = 8). The inner part of the meat was sampled to verify the absence of microorganisms. The meat samples were collected again after 14 days from all 15 samples subjected to meat aging (n = 30). A total of 38 samples were examined from the inner part of whole pieces of pork.

#### 2.2.2. Microbiological Analyses

All samples (10 g) were homogenized with 90 mL of buffered peptone water (BPW; OXOID Ltd., Basingstoke, UK) in a stomacher Star Blender LB 400 (VWR, Radnor, PA, USA), and, further, tenfold dilutions were prepared as required. The microorganisms were determined by culture methods following ISO standards. The aerobic and facultative anaerobic psychrotrophic microorganisms (total viable psychrotrophic count; pTVC) were determined using glucose, tryptone, and yeast extract agar (OXOID Ltd.) [17]. Violet Red Bile Glucose agar (VRBG agar; OXOID Ltd.) incubated at 30 °C for 24 h was used to determine the number of bacteria of the *Enterobacteriaceae* family [18]. Five colonies from each Petri dish were tested for the presence of oxidase (JK Trading, spol. s r. o., Brno, Czech Republic)—negative reaction. The psychrotrophic lactic acid bacteria (pLAB) culture was carried out under anaerobic conditions (AnaeroGen TM; OXOID Ltd.) on de Man, Rogosa, and Sharpe agar (MRS agar; OXOID Ltd.) [19]. Due to the nature of the samples collected, the MRS agar was incubated at a modified temperature (15 °C for seven days). A minimum of five colonies exhibiting distinct morphological characteristics were selected from each Petri dish and subjected to testing for the presence of catalase and oxidase (JK Trading, spol. s r. o.)—both tested with negative reactions. Pseudomonas agar (OXOID Ltd.) incubated at 25 °C for 48 h was used to determine the genus *Pseudomonas* [20]. At least five colonies exhibiting distinct morphological characteristics were selected from each Petri dish and subjected to catalase and oxidase testing (JK Trading, spol. s r. o.)—both tested with positive reactions.

### 2.3. Color and Texture Measurements

For the objective assessment of color and texture, the meat from the neck and loin was deboned and sliced to a thickness of 3 cm. Measurements were taken both in their fresh state and after cooking. The meat slices were cooked using the grill program in a Rational 61E combination steam-convection oven (Rational Czech Republic s.r.o., Prague, Czech Republic) with a core temperature of 70 °C.

Color parameters (lightness, L*; redness, a*; yellowness, b*) were quantified according to the CIEL*a*b* system using a spectrophotometer (Konica Minolta CM-5, Konica Minolta, Japan). The instrument was calibrated with a D65 light source and a standard observer angle of 10°, with a measuring slit of 8 mm and the specular component excluded (SCE). Each sample was measured ten times as soon as possible after preparation. Spectra Magic 3.61 software was used to calculate the parameters.

Instrumental texture measurement of cooked meat was conducted on the day following cooking. Before measurement, the chilled meat samples (2 ± 2 °C) were tempered to room temperature. To ascertain the Warner–Bratzler shear force, 10 × 10 × 20 mm blocks of meat were prepared, and the blocks were positioned at the midpoint of the Warner–Bratzler blade and sheared at a 90-degree angle to muscle fiber orientation. The crosshead speed was set to 80 mm/min. Cylindrical samples (10 mm in height and 12.5 mm in diameter) underwent two cycles of 50% compression with a test speed of 50 mm/min. The hardness of cooked pork neck and loin was analyzed using a two-bite system. The texture of the raw and cooked meat was evaluated using a universal testing machine Instron^®^ 5544, (Instron, Norwood, MA, USA). The mean value of shear force and hardness for each sample was calculated from six partial measurements.

### 2.4. Weight Loss

We also evaluated *weight loss* during the dry-aging process. Samples of pork neck and loin with skin intact were weighed on day 0. This was repeated after 14 days of dry aging. *Weight loss* was calculated by subtracting the weights observed on the days of observation from the initial weights, following the following formula:$$Weight\;loss\;\left(\%\right)=\frac{m_{1}-m_{2}}{m_{1}}\times 100$$
where *m*_1_ is the sample weight (kg) before aging and *m*_2_ weight (kg) after aging.

### 2.5. Statistical Analyses

The microbiological data were transformed into logarithms of the number of colony-forming units (CFU/g) and subjected to a Mann–Whitney U test.

The *weight loss* data were subjected to statistical evaluation via the paired *t*-test, while the color and texture data were evaluated via the two-sample *t*-test.

Significant differences were identified at the 5% level (*p* < 0.05). The statistical analysis was conducted using the UNISTAT^®^ software 6.5 (Unistat Ltd., London, UK).

The microbiological data were transformed into logarithms of the number of colony-forming units (CFU/g) and subjected to an ANOVA. The statistical analysis was conducted using the UNISTAT^®^ software. Significant differences were identified at the 5% level (*p* < 0.05).

The *weight loss* data were subjected to statistical evaluation via the paired *t*-test, while the color and texture data were evaluated via the two-sample *t*-test with unequal variances (Microsoft Office Excel 2016).

## 3. Results and Discussion

### 3.1. Microbiological Parameters

The most prevalent bacterial groups responsible for meat spoilage include *Pseudomonas* spp., lactic acid bacteria, *Brochothrix thermosphacta*, and *Enterobacteriaceae* [21,22]. The pTVC, *Enterobacteriaceae*, pLAB, and *Pseudomonas* spp. were used to investigate the microbial impact on dry-aged pork meat. To provide a comprehensive overview of the assessment of the microbial quality, samples were also collected from inside the individual cuts. The results are presented in Table 1 and Table 2.

The initial pTVC values (day 0) for the surface swabs of the pork and neck together were approximately 3–4 log CFU/cm^2^, which is consistent with the normal total viable count reported for fresh meat [23]. For one out of a total of 60 samples from the surface of the loin, a pTVC of 7.3 log CFU/cm^2^ was detected, reaching the limit for meat spoilage of 7.0 log CFU/cm^2^ [24,25]. There was no statistically significant increase in pTVC during aging (*p* > 0.05). However, on the surfaces along which the meat was cut (both neck and loin), seven samples exhibited pTVC counts exceeding 7.0 log CFU/cm^2^. However, we saw no evidence of surface spoilage of the meat, such as discoloration, slime production, or atypical odor.

Bacteria from the *Enterobacteriaceae* family serve as indicators of a hygienic environment [26], and they were approximately 2 log CFU/cm^2^ on the surface of the skin at the beginning of aging, while values on the lateral sides were below the limit of detection. These initial values are comparable to Augustin & Minvielle’s study [27]. After 14 days of aging, *Enterobacteriaceae* numbers demonstrated no discernible trend and remained stable (*p* > 0.05), except on the skin side of the neck, where they declined below the limit of detection (*p* < 0.05). The observed decline in *Enterobacteriaceae* numbers may be attributed to the drying of the skin surface and the associated reduction in water activity, which represents an unfavorable environment for their survival.

pLAB counts at day 0 on the skin surface were found to be 2.74 log CFU/cm^2^ (neck) or 2.80 log CFU/cm^2^ (loin). They were 1 log CFU/cm^2^ lower on the side sections. After 14 days of aging, the pLAB count demonstrated no discernible trend. The findings of Mikami et al. [28] that lactic acid bacteria are suppressed during the dry-aging process were not confirmed by statistical analysis. However, the results demonstrated a decrease below the limit of detection on the skin, starting from values of 2.74 and 2.80 log CFU/cm^2^.

Similar results were obtained for swabs from the cut surfaces, where initial values were lower by 1 log CFU/cm^2^ than those on the skin surface (see Table 1). There was no statistically significant increase in pLAB counts after 14 days of aging.

Pseudomonads are the most prevalent group of bacteria responsible for meat spoilage under aerobic conditions [22]. The low temperature and air atmosphere during dry aging facilitate the growth of these highly proteolytic bacteria [29,30]. In our study, *Pseudomonas* spp. counts on the skin surface remained stable (*p* < 0.05), likely due to skin drying during aging and the associated reduction in water activity on the surface of the pork neck and loin together. The values ranged from 2.7 log CFU/cm^2^ to 5.72 log CFU/cm^2^ on day 0 and below the detection limit of 4.57 log CFU/cm^2^ after 14 days of aging. Similar findings have been reported by other authors [27,31,32], including Endo et al. [32], who observed a significant decrease in *Pseudomonas* spp. on the skin surface after 20 days of aging, with values returning to baseline levels after a further 20 days of aging.

Results from the cut sides showed that *Pseudomonas* numbers increased by 2–3 log CFU/cm^2^. Here, the skin did not function as a protective barrier. However, the increase was not statistically significant (*p* > 0.05).

The impact of dry aging on the growth of microorganisms was not assessed only on the surface but also in samples extracted from inside the individual meat pieces. We found no statistically significant difference between day 0 and day 14 in the numbers of pTVC, *Enterobacteriaceae*, pLAB, and *Pseudomonas* spp. This can be attributed to the protective function of the skin, which acts as a barrier against the penetration of bacteria into the deeper layers. Furthermore, dry aging removes moisture from the surface layer of the meat, reducing water activity. This, in turn, suppresses the growth of aerobic bacteria and reduces the overall level of contamination [33,34]. The probable cause of bacteria occurrence inside meat is the bacteria migrating into meat via gaps between muscle fibers and endomysia. It would seem that the increasing osmolality of muscle going into rigor, which results from lactic acid formation, causes radial shrinkage of muscle fibers. Fiber shrinkage produces gaps between the contractile elements of cells and the surrounding endomysia. These gaps would offer the most obvious route for bacterial invasion [35,36].

Our approach differs from that of most similar studies that have evaluated the effects of dry aging on pork [37,38,39], which have primarily examined the effects of aging on the physicochemical and sensory parameters of the meat. The objective of our study was more comprehensive in the evaluation of microbiological indicators. The first study to link the physicochemical and sensory properties of aged pork to health requirements was published as late as 2020 [15].

In our experiment, the same portion of the pork carcass was analyzed in all animals. This was carried out immediately after slaughter, at the slaughterhouse, in a dedicated area where sampling was conducted. The aging process occurred in the aging chamber of the holding facility alongside other meat intended for sale in the market. The handling of the samples before and after sampling was comparable on each sampling day and was carried out with the assistance of the slaughterhouse personnel who prepared the meat samples. It can therefore be concluded that these results accurately take into account conditions that prevail during real-life operations.

### 3.2. Pork Color

The results of the color measurements are shown in Table 3. There was no significant difference throughout dry aging in the lightness (L*) and redness (a*) values of pork neck or loin either in the raw or cooked meat. There was a significant decrease (*p* < 0.05) in yellowness (b*) in cooked, dry-aged loins. In contrast, raw neck meat showed significantly higher b* (*p* < 0.05) after aging than at the beginning of the experiment. Previous studies have shown that pork loins show an increase in L* values (*p* < 0.05) during dry aging due to moisture evaporation, and the consequent reduced light reflection [39]. Juárez et al. [40] related the increase in L* values (*p* < 0.001) to an increase in the proportion of oxymyoglobin and a decrease in myoglobin content. It has been reported that the extent and rate of oxygen diffusion to the meat surface increases during dry aging as oxygen-consuming enzymes are gradually inactivated. Similar to our study, Hwang et al. [41] found only a slight increase (*p* > 0.05) in a* values during dry aging of pork belly and shoulder. While the work of some authors [39,42] reports non-significant changes in b* values, Hwang et al. and Tikk et al. [41,43] reported a significant increase in b* values during pork aging. The color parameters of the pork samples were also influenced by sample size and the processing method. In our case, the surface fat and skin impeded the moisture evaporation and oxidation of a portion of the muscle surface. The studies mentioned above report results based on pork samples devoid of fat and skin [39,40,41,42], or from whole carcasses [43]. It is important to note that the values of b* may fluctuate during cooking. The observed discrepancy in b* values for cooked loins may be attributed to the varying extent of fat oxidation [44]. Maillard reaction products form on the surface during the heat treatment, resulting in the red color changing to white, gray, brown, or black. Nevertheless, the b* values on the surface of the meat were not found to be statistically significant (*p* > 0.60) [45].

### 3.3. Warner–Bratzler Shear Force and Hardness

Although previous studies have reported a reduction in Warner–Bratzler shear force (WBSF) values during the dry aging of pork [40,41,46], this study found no difference in WBSF values between fresh and aged pork neck and loin, both in raw or cooked meat (Table 3). These results are consistent with those of Xu et al. [47], who found no difference in WBSF after 12 days of pork aging (*p* = 0.42). WBSF is frequently employed as a measure of meat tenderness in numerous studies. However, the correlation between WBSF and sensory evaluations of tenderness has been found to range from very high to very low, with correlation coefficients ranging from −0.914 to −0.001, respectively. One potential explanation for the observed low correlation is the poor performance of sensory panels. The underlying factors determining variation in meat tenderness include connective tissue cross-linking, myofibrillar integrity, sarcomere length, intramuscular fat, and protein denaturation during cooking [48]. Similar to WBSF, there was no significant difference (*p* > 0.05) in hardness between fresh and aged loins after cooking (Table 3). Similarly, Gu et al. and Xu et al. [46,47] also observed no change in pork hardness after 14 and 12 days of aging, respectively (*p* = 0.49 and *p* = 0.82). Lee et al. [33] reported significant changes (*p* < 0.05) in the hardness of pork loin after 40 days of dry aging. The observed differences in the hardness of fresh and aged neck meat following heat treatment may be influenced by variations in the proportions of connective tissue and fat, given that pork neck is composed of multiple muscles.

### 3.4. Weight Loss

Following 14 days of dry aging of pork neck and loin, a significant (*p* < 0.001) *weight loss* was observed, ranging from 5.1% to 8.3%. The observed values of *weight loss* during dry aging are influenced by the conditions within the aging chamber, as well as the treatment and size of the meat samples. In a 2011 study, Juárez et al. [40] evaluated *weight loss* in pork loins without skin and fat. This significant loss (*p* < 0.05) was higher after 14 days than in our study (9.8%), and it was not compensated by significant improvements in pork flavor or tenderness attributes assessed by trained panelists using the standard guidelines. A further significant *weight loss* (41.19%; *p* < 0.05) was observed after 18 days of dry aging by Vinauskienė et al. [49] in 200 g slices of pork loin that were 2.8 ± 0.2 cm thick. The *weight loss* was found to be influenced by the type of aging (dry versus wet) and the aging period (*p* < 0.001) [40]. Consequently, the dry aging of pork is recommended only for meat in its intact form, that is, with bone and skin.

## 4. Conclusions

The aging of meat is associated with enzymatic (proteolysis), physical (*weight loss*), physico-chemical (oxidation), or microbiological changes that can improve the sensory properties of meat (texture, taste), but also deteriorate them (spoilage due to the growth of microorganisms). While aging is a common procedure for improving the culinary properties of beef, it is still marginally used for pork.

To reduce economic losses during aging (*weight loss* or microbiological spoilage), it is necessary to choose a suitable method of dry or wet aging. The technology of swine slaughtering, combined with scalding and leaving the skin as a natural protection of the carcass, provides a basic prerequisite for the use of the dry aging of meat. The main weakness of dry aging beef is the risk of *weight loss* due to water evaporation and the need to cut off the surface-dried and -oxidized layers of the meat. If we leave the natural cover of the skin free of bristles, the pork is perfectly protected from adverse microbial changes and the oxidation and drying of the meat surface. As pork generally shows an earlier onset of spoilage compared to beef, wet aging of pork is not recommended. In order to improve the texture of the meat, it is necessary for the pork to age for more than 10 days. Handling the meat before vacuum packaging increases the risk of cross-contamination of the meat, and, according to our previous experience, the wet aging of pork after 14 days can already be accompanied by signs of spoilage. However, this will not happen with properly selected dry aging, which was confirmed by this study. While preserving the natural surface covered by skin, the *weight loss* of meat with bones was a maximum of 8%, without significant other waste.

The number of studies with clearly defined conclusions on the dry aging of pork meat is still deficient. The combination of data from instrumental measurements and microbial analysis indicates that 14 days of dry aging is sufficient for pork aging and does not significantly affect the microbiological profile. Nevertheless, it is necessary to optimize the dry aging time for various types of meat to achieve the desired characteristics and to prevent hygiene issues.

## Figures and Tables

**Figure 1 foods-13-03037-f001:**
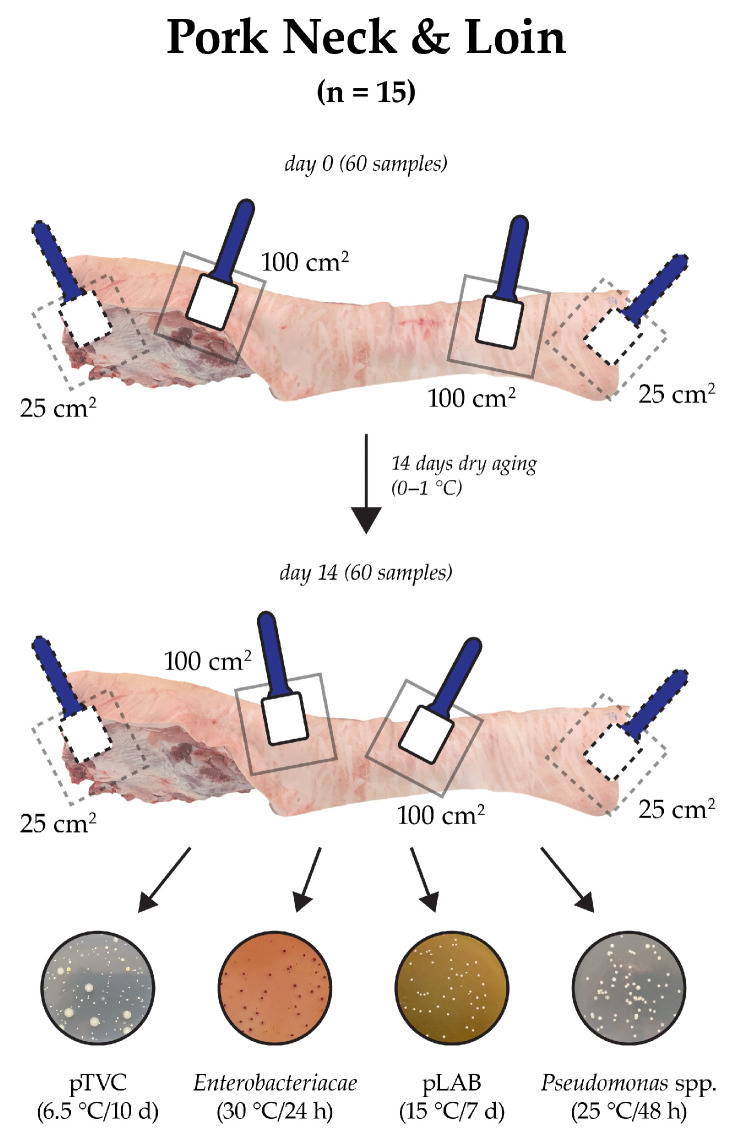
Scheme of sampling from the surface of pork meat for microbiological analysis (pTVC: total viable psychrotrophic count; pLAB: psychrotrophic lactic acid bacteria).

**Table 1 foods-13-03037-t001:** Results of the psychrotrophic TVC, *Enterobacteriaceae,* psychrotrophic LAB, and *Pseudomonas* spp. from skin surfaces and cut surfaces on day 0 (**A**) and 14 days after slaughter (**B**). Results are presented in log CFU/cm^2^.

Samples		pTVC		*Enterobacteriaceae*		pLAB		*Pseudomonas* spp.	
		Median Min; Max	*p*-Value	Median Min; Max	*p*-Value	Median Min; Max	*p*-Value	Median Min; Max	*p*-Value
Skin surface of neck	A	4.63 1.98; 6.11	>0.05	2.26 1.08; 3.08	<0.05	2.74 <1.00; 5.20	>0.05	3.67 2.70; 5.11	>0.05
	B	3.95 <1.00; 6.62		0.00 0.00; 0.00		<1.00 <1.00; 4.23		2.57 <1.00; 4.57	
Skin surface of loin	A	4.85 3.20; 7.34	>0.05	2.63 1.30; 4.46	>0.05	2.80 1.30; 4.72	>0.05	3.94 3.15; 5.72	>0.05
	B	4.20 3.23; 5.56		0.00 0.00; 1.49		<1.00 <1.00; 4.88		3.00 <1.00; 4.48	
Cut surface of neck	A	3.42 2.08; 6.53	>0.05	<0.60 <0.60; 2.83	>0.05	1.60 <1.60; 3.26	>0.05	2.98 <1.60; 4.18	>0.05
	B	5.83 <2.60; 9.48		<0.60 <0.60; 2.88		2.30 <1.60; 5.58		5.54 <1.60; 7.59	
Cut surface of loin	A	3.36 <1.00; 6.72	>0.05	<0.60 <0.60; 2.78	>0.05	1.30 <1.60; 2.68	>0.05	2.78 <1.60; 3.58	>0.05
	B	6.02 1.90; 8.53		<0.60 <0.60; 3.78		2.60 <1.60; 6.49		5.41 <1.60; 6.9	

**Table 2 foods-13-03037-t002:** Results of psychrotrophic TVC, *Enterobacteriaceae,* psychrotrophic LAB, and *Pseudomonas* spp. from inside the meat on day 0 (**A**) and 14 days after slaughter (**B**). Results are presented in log KTJ/g; n = 30.

Sample	pTVC		*Enterobacteriaceae*		pLAB		*Pseudomonas* spp.	
	A	B	A	B	A	B	A	B
Median	2.26	4.23	<1.00	<1.00	<2.00	<1.00	<2.00	3.65
Min	<1.00	1.88	<1.00	<1.00	<2.00	<2.00	<2.00	<1.00
Max	3.11	6.59	<1.00	1.88	<2.00	4.26	3.08	6.63
*p*-value	>0.05		>0.05		>0.05		>0.05	

**Table 3 foods-13-03037-t003:** Color and texture properties of raw and heat-treated fresh and dry-aged pork loin and neck (n = 76); mean ± standard deviation.

Cut	Character	L*	a*	b*	Shear Force	Hardness
Loin	FR	52.25 ± 4.62	0.42 ± 1.26	9.45 ± 1.58	26.98 ± 8.66	-
	AR	51.93 ± 2.99	1.11 ± 1.64	10.25 ± 1.62	31.60 ± 6.18	-
	FC	71.01 ± 2.84	2.70 ± 1.06	18.80 ± 1.51 ^a^	51.37 ± 20.60	21.69 ± 11.20
	AC	69.59 ± 2.20	2.40 ± 0.68	16.18 ± 1.17 ^b^	35.43 ± 6.88	33.11 ± 7.75
Cut	Character	L*	A*	B*	Shear Force	Hardness
Neck	FR	41.82 ± 1.62	6.00 ± 0.98	8.17 ± 0.99 ^a^	25.19 ± 6.47	-
	AR	43.15 ± 3.40	6.87 ± 1.58	9.71 ± 1.87 ^b^	24.78 ± 5.07	-
	FC	53.72 ± 0.73	5.88 ± 0.26	16.64 ± 1.57	30.46 ± 5.75	9.53 ± 2.43 ^a^
	AC	53.49 ± 3.15	6.04 ± 1.04	16.10 ± 1.87	26.82 ± 9.10	16.02 ± 2.76 ^b^

FR = Fresh Raw; AR = Aged Raw; FC = Fresh Cooked; AC = Aged Cooked.^a,b^: values with different superscripts in the same column within the same parameter and the same meat character (raw/cooked) differ significantly (*p* < 0.05).

## Data Availability

The original contributions presented in this study are included in the article, and further inquiries can be directed to the corresponding author.

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
