# Peer review of "Effect of Dry Aging of Pork on Microbiological Quality and Instrumental Characteristics"

_foods, 2024, doi:10.3390/foods13193037_

Round 1
Reviewer 1 Report
Comments and Suggestions for Authors
The work is very topical and interesting. In attachment, you will find some corrections or it is better to say some suggestions to make the work even better.
All the best,

Comments on the Quality of English Language
English is not my native language, so I don't think I'm competent to talk about it.
Author Response
RESPONSE TO REVIEWER #1
We thank the reviewer for the valuable comments.
It is better to put this part of introduction about tenderness in discussion.
The mentioned sentences were partly modified.
P has to be small letter "p" and italic.
It has been corrected.
Please add information about weight and the age of pigs.
Information has been added.
Why only 8 samples? Why not all 15?
These samples only served to verify the absence of microorganisms in the depth of whole pieces of meat after slaughter. Therefore, sampling was limited to 4 neck samples and 4 loin samples from the right halves of the carcasses.
Incubation of Enterobacteriaceae should be at 37 °C.
In order to detect psychrotrophic bacteria of the Enterobacteriaceae family, a temperature of 30 °C was chosen.
It is not necessary to explain the confirmatory tests since you have referred to the ISO method.
For easier orientation, it is customary to briefly state the procedures, so we followed this practice and provided the information.
Did you take into account the 10 plus 10 g for microbiology? Please add informations about that.
Of course, the weight was determined before sampling.

Reviewer 2 Report
Comments and Suggestions for Authors
The work studies the maturation process of pork meat comparing traditional processes with a dry process. Microbiological and sensory aspects are measured.
Following are some aspects to be improved in the paper:
l5 Veterinary sciences Brno, Czech Republic?
Improve the wording l14-l16
In l18 werner bratzel is not the analysis but rather the probe used, revise.
The wording of the introduction is a bit rough, revise the wording to make it more fluent.
If in l50 there is mention of time milestones, could the idea/concept be developed a little more from 1970 to 2024...?
It would be interesting to talk about dry and wet aging in a balanced way and make some comparison of the two by means of a table that considers main characteristics, advantages and disadvantages of both processes.
In l58 the time range is established, shouldn't it be something more general like the evolution in time and then in the body of the paper the time is mentioned, specifically the 14 days...?
In 2.1 If the work starts from the sacrifice, some kind of bioethics certification should be presented?
The characteristics of the pigs such as breed, stage of development and any other condition that would allow standardization of the experiment are not adequately established.
In l72, the aging chamber is mentioned, but there is no mention of information such as chamber humidity, meat humidity?
In the experimental part there is information that is repeated in l64 l77 for example. It is necessary to review the whole document to avoid repeating information.
In some parts the experiments are not completely clear, it would be interesting to add some kind of schematic diagram to facilitate a quick understanding of the experiments.
2.2.1 could be reduced and written more synthetically and clearly, as it is somewhat convoluted.
In l105, ISO standards are mentioned but the specific numbers that complete the identification are not mentioned.
l125 uncooked can be replaced by fresh?
In l134 it is necessary to justify why measurements are made after one day of cooking.
Is l165-172 really necessary?
In relation to the microbiological results, is there any limit or normative recommendation that the authors consider necessary?
l184 I don't understand what you are referring to.
L188 Is the water and moisture activity of the samples measured anywhere?
The conclusion should be completely rewritten with emphasis on the new knowledge generated, main findings, its usefulness, possible applications and future work derived from this knowledge.
Author Response
RESPONSE TO REVIEWER #2
We thank the reviewer for the valuable comments.
l5 Veterinary sciences Brno, Czech Republic?
The workplace name is correct.
Improve the wording l14-l16
The text has been modified according to the requirements.
In l18 werner bratzel is not the analysis but rather the probe used, revise.
The term Warner-Bratzler has been removed.
The wording of the introduction is a bit rough, revise the wording to make it more fluent.
The Introduction chapter has been completely revised.
If in l50 there is mention of time milestones, could the idea/concept be developed a little more from 1970 to 2024...?
Edited as part of revision of the Introduction chapter. But we did not want to comprehensively describe the details of meat aging, as this is not a review-type article.
It would be interesting to talk about dry and wet aging in a balanced way and make some comparison of the two by means of a table that considers main characteristics, advantages and disadvantages of both processes.
We did not want to comprehensively describe the details of meat aging, as this is not a review-type article.
In l58 the time range is established, shouldn't it be something more general like the evolution in time and then in the body of the paper the time is mentioned, specifically the 14 days...?
It has been changed as requested.
In 2.1 If the work starts from the sacrifice, some kind of bioethics certification should be presented?
Pigs were slaughtered in commercial slaughterhouse in accordance with European legislation (Regulation (EC) 1099/2009) and as part of normal operations. The slaughter was carried out in accordance with the applicable legislation. Commercial slaughterhouse was approved by the competent authority and the operations are under permanent veterinary supervision. There is no need to submit any additional documents for approval by the ethics committee.
The characteristics of the pigs such as breed, stage of development and any other condition that would allow standardization of the experiment are not adequately established.
The required information was added in the text.
In l72, the aging chamber is mentioned, but there is no mention of information such as chamber humidity, meat humidity?
Information on air humidity has been added. Chemical analyzes of the meat composition were not performed as part of the experiment.
In the experimental part there is information that is repeated in l64 l77 for example. It is necessary to review the whole document to avoid repeating information.
Duplicate information has been removed from the text.
In some parts the experiments are not completely clear, it would be interesting to add some kind of schematic diagram to facilitate a quick understanding of the experiments.
A scheme of sampling and further microbiological processing was inserted into the manuscript.
2.2.1 could be reduced and written more synthetically and clearly, as it is somewhat convoluted.
Chapter 2.2.1 has been revised according to requirements.
In l105, ISO standards are mentioned but the specific numbers that complete the identification are not mentioned.
The link to the full text of the ISO standards is given in References.
l125 uncooked can be replaced by fresh?
It has been revised according to requirements.
In l134 it is necessary to justify why measurements are made after one day of cooking.
Studies (e.g. Colle et al., 2015; http://dx.doi.org/10.1016/j.meatsci.2015.06.013) mostly measure shear force using the Warner-Bratzler method only on the second day after cooking.
Is l165-172 really necessary?
Part of the discussion on the microbial quality of pork, we recommend keeping it. However, the text has been reduced
In relation to the microbiological results, is there any limit or normative recommendation that the authors consider necessary?
A level of 7.0 log CFU/cm2 is generally considered to be the limit of microbial spoilage of meat, has been revised in the text.
l184 I don't understand what you are referring to.
Revised as required.
L188 Is the water and moisture activity of the samples measured anywhere?
The water activity values ​​of meat and pork skin were not measured due to the difficulty of processing such an inhomogeneous material, which is pork skin (surface epidermis and inner layer of the joint with a proportion of fat).
The conclusion should be completely rewritten with emphasis on the new knowledge generated, main findings, its usefulness, possible applications and future work derived from this knowledge.
The conclusion has been completely revised.
